# Flood Risk Analysis for Cascade Dam Systems: A Case Study in the Dadu River Basin in China

**Wenjun Cai [1,*], Xueping Zhu [1], Anbang Peng [2,3], Xueni Wang [1] and Zhe Fan [4]**

[1] College of Water Resources Science and Engineering, Taiyuan University of Technology, Taiyuan 030024, China

[2] State Key Laboratory of Hydrology-Water Resources and Hydraulic Engineering, Nanjing Hydraulic Research Institute, Nanjing 210029, China

[3] Hydrology and Water Resources Department, Nanjing Hydraulic Research Institute, Nanjing 210029, China

[4] China Institute of Water Resources and Hydropower Research, Beijing 100038, China

**\*** Correspondence: caiwenjun62620@163.com

**Abstract:** To quantify the flood risks in cascade dam systems, it is critical to analyze the risk factors and potential breaking failure paths. In this study, Bayesian networks (BNs) were applied to create a flood risk analysis model for a cascade dam system. Expert experiment, historical data, and computational formulas were employed to estimate the prior probability and original conditional probability tables (CPTs) in the BN model; sensitivity analysis was used to ensure the original continuous breaking failure path in the system. To avoid the possible misperceptions of the probability of a certain event, Dam Breach Analysis Model (DB-IWHR) 2014 software and the flood regulation method were used to simulate the dam breaking progress. The posteriori continuous breaking failure paths were obtained, and then the original CPTs were refined based on the new evidence. The proposed method was applied to the Bala-Busigou-Shuangjiangkou (BL-BSG-SJK), which is located upstream of the Dadu River basin in China. The results show that three continuous breaking failure paths could be identified in the researched cascade dam system. A new BN model was created to determine the failure probability of the cascade dam system under the three continuous breaking failure paths. This analytical method may also be useful for other similar cases.

**Keywords:** cascade dam system; flood risk analysis; dam breaking

## 1. Introduction

In China, there are almost 98,000 dams with a combined storage capacity of $9.32 \times 10^9$ m$^3$ [1]. To achieve goals like flood control, hydroelectric power, irrigation, and navigation, several large-scale cascade dam systems have been constructed in the Yangtze River, Jinsha, Yellow River, Yalong River, Lancang River, Wujiang, Red River, and Dadu River Basin [2]. The statistical data reveal that more than 95% of these dams are embankment dams in China. From 1954 to 2013, approximately 3523 dam failure accidents occurred, resulting in fatalities and economic losses [3–5]. Among the diverse natural hazards, flooding is the most important risk factor affecting dam breaking.

Flooding is the most disastrous natural hazard for the basin, and floods are transferred to the cascade dam system, like the domino effect [6]. Therefore, flood risk analysis for cascade dam systems is important. Chen et al. developed a risk-based model for real-time flood control operation of dams under emergency and uncertain conditions [7]. Due to the properties of the engineering system, Bayesian networks (BNs) are employed to quantify the complex relational dependencies using Bayes' theorem. BNs are a type of probability graphical model that can accurately predict one event's probability by combining historical data and expert experience [8]. Due to the flexible

structure and cause–effect inference engine, BNs are a promising tool for risk analysis in complex systems [9]. BNs have been extensively applied for the analysis of the failure probability of gas pipelines [10,11], reliability estimation of system functioning [12,13], and risk assessment for reservoirs with respect to water quality and public health [14].

Application of BNs for flood risk estimation of cascade dam systems is still in the preliminary phase. For instance, Li et al. used a BN and stochastic Monte Carlo to analyze the dam breaking risks of two reservoirs under floods and landslide surge [15]. During the flood risk analysis, it is crucial to identify the risk factors and continuous breaking failure paths of cascade dam systems. On the basis of prior probability in the original BN model, sensitivity analysis can be used to rank the major risk factors for failure events of the system [16,17]. Considering the possible misperceptions of the expert experiment, the risk factors and failure path identified by the original prior probability and conditional probability tables (CPTs) cannot truly reflect the cascade dam system situation. Therefore, the original prior probability and CPTs need to be updated after the analysis of continuous breaking conditions.

During the simulation of the dam breaking process, the prediction accuracy of flood release is important for the breach of artificial or natural dams [18]. In general, a combination of the hydraulic and the geological method is used to create the dam breaking analysis model; the flood release routing for dam breaking is often ensured by the broad crested weir flow formula [19]. Several analytical models can be used to simulate the dam breaking process, such as Hydrologic Modeling System-River Analysis System (HEC-RAS) [20,21] and MIKE 11 [22]. Wahl and Zhu et al. pointed out some deficiencies in the former publications, which mainly included the prediction uncertainties of the dam breaking process [23,24]. For instance, the empirical parametric models always overvalue the peak outflow. According to Liu et al. the simulated peak outflow is 4600 $m^3$/s, which is eight times as large as practical discharge [25]. To solve this problem, Chen et al. developed a simple numerical algorithm to improve the existing dam breaking simulation method, which can avoid iteration increases each time [26]. The proposed algorithm, DB-IWHR, can be run in Excel 2014. The improvements of the model are able to reduce the sensitivity of the input parameters for dam breaking analysis [19]. The hydraulic details and lateral enlargement of dam breaking progress have been reported in the former two publications [19,26]. According to Zhou et al. and Zhang et al., DB-IWHR 2014 can be used to analyze the risks for Hongshiyan Barrier Lake and Dadu River cascade dam [27,28]. Based on the previous studies about the prediction of the dam breaking process, DB-IWHR 2014 was employed to simulate dam breaking progression in this study.

In this study, the failure probability for the cascade dam system was estimated through a BN combined with DB-IWHR 2014. First, the BN model was created to study the logical dependency relationship of the cascade dam system. The prior probability and CPTs of each node were obtained from historical data, computational formulas, and expert experience. In the BNs model, the sensitivity was analyzed to determine the original continuous break failure path. Next, DB-IWHR 2014 was applied to analyze the reasonability of the original failure path, which can avoid the use of subjective factors to some extent. Posteriori continuous break failure paths can be certified by DB-IWHR 2014 under the different sates of each node. After the posteriori paths are determined, the original CPTs are sequentially refined. Finally, based on the redefined CPTs and new evidence, a new BN model was created to calculate the system failure probability to assess the effects of each dam on the system safety. The proposed method was applied to the Bala–Busigou–Shuangjiangkou (BL–BSG–SJK) cascade dam system, which is located in the Dadu River Basin in China. On the basis of the application, three reasonable continuous breaking failure paths were identified, and the cascade dam system failure probabilities of the determined continuous breaking failure paths were calculated by the refined BNs model.

## 2. Study Area

Dadu River, the largest tributary of the Ming River, is located in Sichuan Province, China, as shown in Figure 1. The total length of its main stream is 1062 km and its catchment area is 77,400 $km^2$. Dadu River is geographically positioned between 99°42′ and 103°48′ E and 28°15′ and 33°33′N

within the transitional zone from the southwest area of the Tibetan plateau to the Sichuan catchment. Above the SJK dam, the typical alpine valley region and fast-flowing river have an average bed slope of approximately 6.2‰.

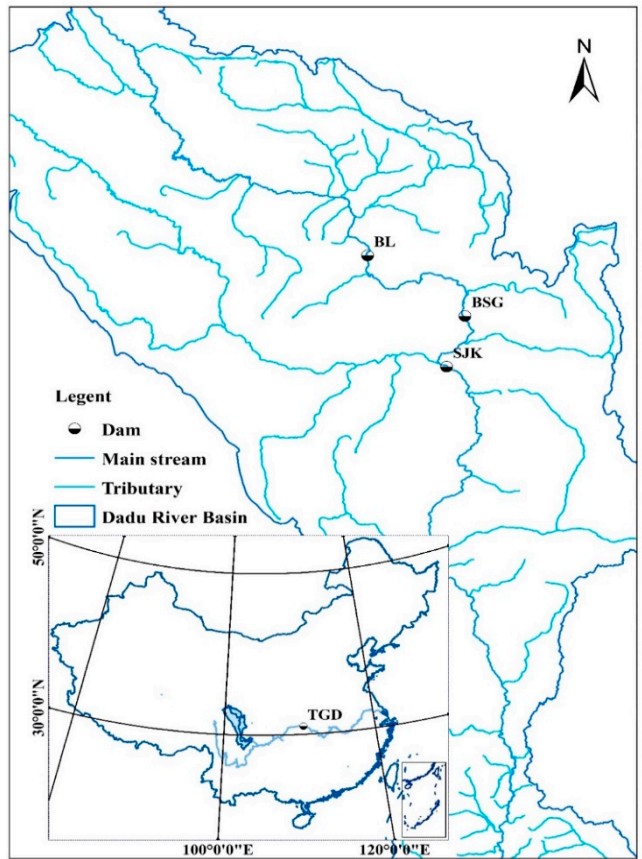

**Figure 1.** Location of Dadu River Basin and Bala–Busigou–Shuangjiangkou (BL–BSG–SJK) cascade dam system.

Dadu River has rich water resources. According to the Dadu River hydropower program (Power China Chengdu engineering corporation, 2013), there are 22 dams constructed or under construction in the Dadu River Basin, with a total installed capacity of $2.3 \times 10^7$ KW. For a controlled engineering project in Dadu River Basin, the SJK dam is not only in charge of flood control for the upstream Dadu River Basin, but also provides flood control pressure relief for the Three Gorges Dam (TGD) in the south of China. Hence, it is necessary to analyze the safety of SJK. We selected two upstream dams adjacent to SJK so the three dams included the BL–BSG–SJK cascade dam system.

In this paper, the BL–BSG–SJK cascade dam system was selected as a research object. The checking standard of BL–BSG is a 5000-year return period, and the checking standard of SJK is the probable maximum flood (PMF). The main parameters of the three dams are listed in Table 1. BL, BSG, and SJK are all embankment dams, according to total capacity and maximum dam height. The three dams are classified as Type I according to the rank of the water and hydropower project and flood protection criteria (SL252-2017) in China [29].

**Table 1.** Main parameters of cascade dams.

| Item | BL | BSG | SJK |
|---|---|---|---|
| **Dam Type** | **Embankment Dam** | **Embankment Dam** | **Embankment Dam** |
| Elevation of dam crest | 2925.00 m | 2608.00 m | 2507.70 m |
| Maximum height of dam | 142.00 m | 133.00 m | 314.00 m |
| Check flood water level | 2922.10 m | 2603.32 m | 2504.42 m |
| Normal flood water level | 2902.00 m | 2603.00 m | 2500.00 m |
| Dead water level | 2900.00 m | 2600.00 m | 2420.00 m |
| Total storage | $1.38 \times 10^8$ m³ | $2.48 \times 10^8$ m³ | $28.97 \times 10^8$ m³ |
| Flood control capacity | \ | \ | $6.63 \times 10^8$ m³ |
| Maximum discharge | 3669.00 m³/s | 4230.00 m³/s | 8101.00 m³/s |

## 3. Methods

### 3.1. Dam Breach Analysis Model

DB-IWHR 2014 [26], as a user-friendly, transparent, and robust tool, can help the researchers predict the dam breaking process. The main aspects involved in the dam breach analysis model are briefly stated below.

By equating the discharge through the breach, the dam breach floods have been modeled as a broad-crested weir, whereas the water storage in the reservoirs is reduced in a unified time. In the dam breaking analysis model, the equation is presented as follows:

$$Q = CB(H-z)^{3/2} = \frac{\Delta W}{\Delta H}\frac{\Delta H}{\Delta t} + q \tag{1}$$

where $C$ is the discharge coefficient, and its theoretical value ranges from 1.3 to 1.7 [30]; $B$ is the width of the weir; $Q$ is the flow discharge for the loss of water storage in the reservoir; $q$ is the natural inflow into the reservoir; $W$ is the water storage capacity of the reservoir; $z$ is the elevation of breach bed; $t$ is the time; and $H$ is the function of the water level. To ensure the continuous positive increase in $H$, $\Delta H$ is calculated from $t$ to $t + \Delta t$:

$$\Delta H = H(t) - H(t+\Delta t) \tag{2}$$

The soil erosion model is the relationship curve of the shear stress and the soil erosion rate. In many studies, an exponential expression has been used for non-cohesive materials [31].

$$\dot{z} = \Phi(\tau) = \frac{v}{a+bv} \tag{3}$$

where $\dot{z}$ is the erosion rate in $10^{-3}$ mm/s, $\tau$ is the shear stress in Pa, $a$ and $b$ are the experience coefficients, and $v$ is the shear stress regarding its critical component.

$$v = k(\tau - \tau_c) \tag{4}$$

where $k$ is the unit conversion factor that allows $\dot{z}$ to approach its asymptote $\dot{z}_{ult}$ within the working range of $\tau$. In addition, DB-IWHR was proposed to be incorporated with the hyperbolic curve [26]. An asymptote is represented by $\dot{z}_{ult} = 1/b$ if $\tau - \tau_c$ approaches an infinite value (Figure 2).

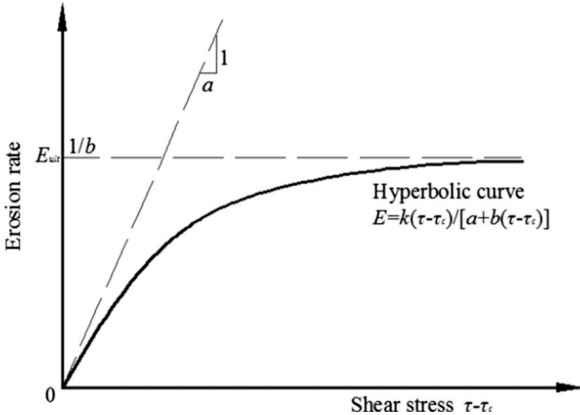

**Figure 2.** The hyperbolic curve of soil erosion rate.

In this model, the asymptote $1/b$ represents the maximum possible erosion ratio, $1/a$ represents the tangent of the curve at the incipient stress. To solve Equations (1) and (3), a new method that is incorporated with DB-IWHR 2014 was used to calculate the incipient velocity $V_0$ at an interval $\Delta V$. [26]. The robust and convenient dam breach analysis model was calculated in a spreadsheet in DB-IWHR 2014 that can be downloaded from the website [32].

*3.2. BN Model*

The BN method is based on the Bayesian formula. The directed acyclic graph (DAG) represents the random variables and directed arcs to imply the causal relationship of the parent and child nodes. In BNs, the DAG and probability parameters can be quantitatively observed in the model [17,323,334].

In BNs, the primary advantage is that the probability distribution function in the graph can be easily updated to reflect the changes in the joint distribution. The Bayesian formula is:

$$P\left(X_1/X_2\right) = \frac{P\left(X_2/X_1\right) \times P\left(X_1\right)}{P\left(X_2\right)} \tag{5}$$

where $P\left(X_1\right)$ is the prior probability of $X_1$, $P\left(X_2\right)$ is the prior probability of $X_2$, $P\left(X_1/X_2\right)$ is the posterior probability of $X_1$ under the condition of a known event $X_2$, and $P\left(X_2/X_1\right)$ is the posteriori probability of $X_2$ under the condition of a known event $X_1$.

BNs are composed of many successor and predecessor nodes, representing a set of random variables ($X_1, X_2, \ldots, X_n$) based on the conditional independence and chain rule. The arcs and nodes show the structural dependency among the variables in the model; the joint distribution of the nodes can be calculated as a product of the conditional probability functions:

$$P\left(X_1, X_2, \ldots X_n\right) = \prod_{i=1}^{n} P\left(X_i/pa\left(X_i\right)\right) \tag{6}$$

where $pa\left(X_i\right)$ is a set of successor nodes of $X_i$, with $i$ = 1, …, $n$, and $P\left(X_i/pa\left(X_i\right)\right)$ is the conditional probability tables (CPTs), if no predecessor is taken as probability tables (PTs), and linked with each node $X_i$.

*3.3. Risk Analysis of Cascade Dam System*

In this study, a BN inference process was employed to create a risk analysis framework for the cascade dam system. A flow diagram of the proposed method is illustrated in Figure 3, and the framework is explained step by step below.

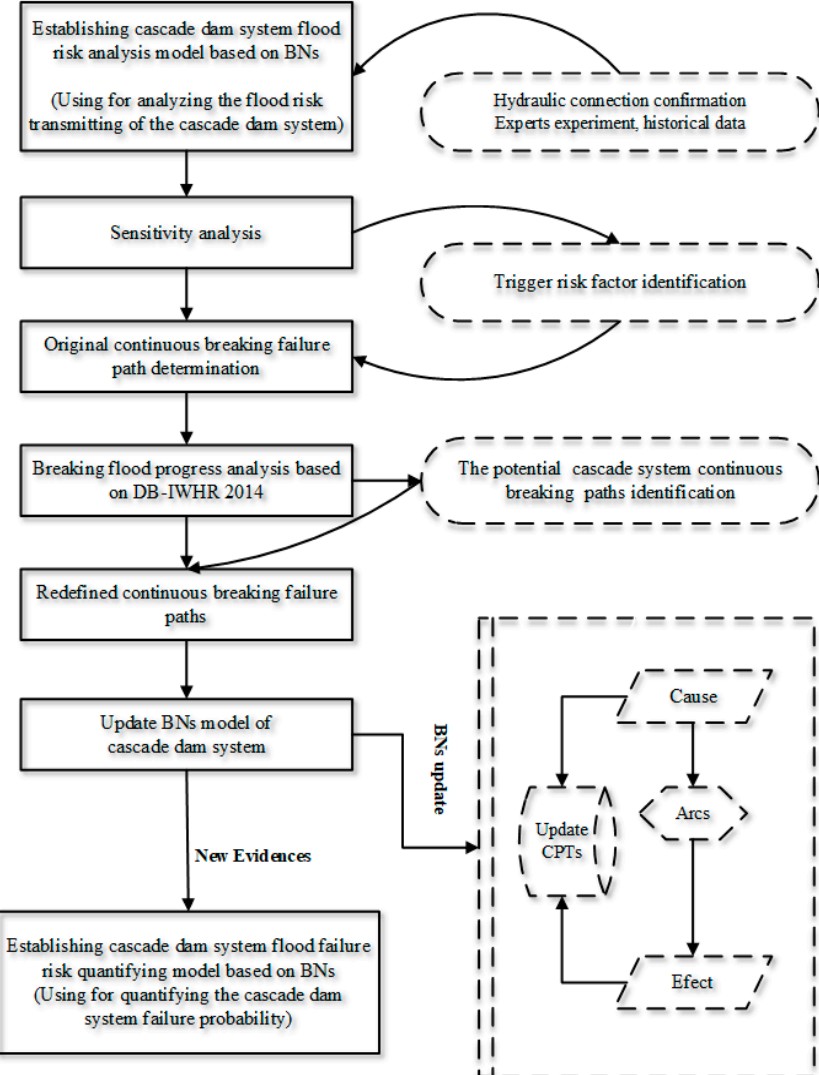

**Figure 3.** Risk analysis flow path of cascade dam system.

Step 1: According to the hydraulic relationship, one BNs model was created to perform the risk transmitting progress of the cascade dam system. In this study, it was necessary to use historical data, computational formulas, and expert experience of the relative information like experiments, simulations, analytical models, and similar previous studies [6,15,26–28,35]. In the BN model, the sensitivity mode was used to obtain the risk factors and original continuous breaking failure path.

Step 2: Step 1 shows that the identified method of the original failure path has to rely on the historical data analysis and expert experience. To obtain more comprehensible and accurate continuous breaking failure paths in the cascade dam system, DB-IWHR 2014 was employed to simulate the dam breaking process.

Step 3: On the basis of simulation results from Step 2, the potential failure paths were identified. For instance, after "BL-Overtopping" occurs, the breaking flood routes downstream and encounters the natural flood, which results in the downstream dam breaking. Eventually, the more reasonable failure paths are identified. The CPTs of the BN model were updated with regard to the new continuous failure paths.

Step 4: In this step, a new BNs model is created to determine the system failure probability for each potential failure path to highlight the effect of each dam on the whole system. The inference

function is achieved using the Bayesian formula. The prior information of this model is obtained from Step 3. As long as new evidence is offered in the directed acyclic graph, the information is transmitted in the model. In the end, the system failure probability is calculated.

## 4. Results and Discussion

### 4.1. Establishment of BNs Model

#### 4.1.1. Model Construction

In this study, the flood risks of the cascade dam system were analyzed using the BN inference process. Due to extreme flooding, the overtopping of embankment dams is the main reason for dam breaking [15,35]. In the BL–BSG–SJK cascade dam system (Figure 4), upstream natural floods and breaking floods are the main factors triggering dam breaks downstream, which are considered the cause of dam overtopping. The nodes and arcs of the DAG indicate the relevant variables and dependencies; the terminating arrow of the arcs reflects the logical relationship of the nodes, which is shown in Figure 5.

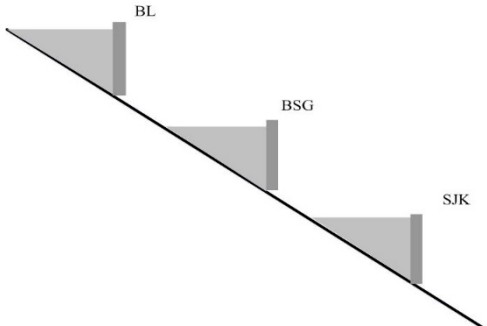

**Figure 4.** Cascade arrangement situation of BL-BSG-SJK.

In this study, the commercial software Hugin 8.5 (Gasværksvej5.DK-9000, Aalborg. Denmark) was applied to establish the BNs model, which can be downloaded on the website [36]. The conceptual model in Figure 5 depicts the application in the artificial intelligence field, which offers convenience for the calculation of model parameters.

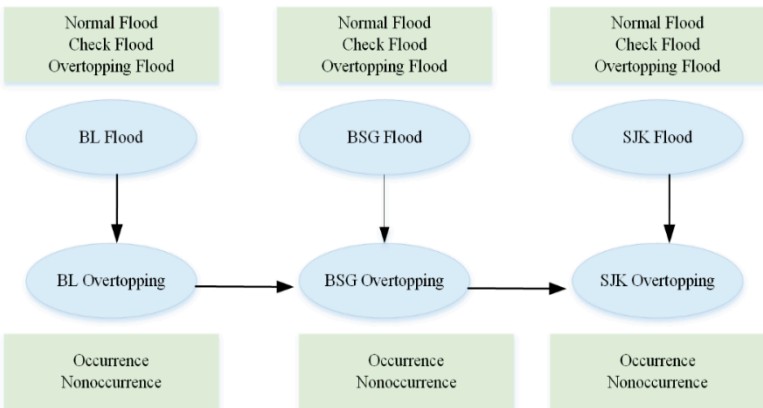

**Figure 5.** The simple mode of connection reservoirs based on the Bayesian network (BN) model.

To make the BNs model more rational and accurate, historical data, experts' experience, and computational formulas are combined to calculate the parameter probability of the nodes. The inference function of the Bayesian formula is composed of conditional probability to determine the state of each node and the probability between child and parent nodes in a CPT. The calculation method is explained as follows:

(1)　Prior probability of each "flood" node (Table 2): This node has three states—normal flood, check flood, and overtopping flood. The probability of each state is determined using the historical data of each dam flood control standard.

**Table 2.** The prior probability of each dam flood node.

| Items | Normal Flood | Check Flood | Overtopping Flood |
|---|---|---|---|
| BL Flood | 0.9997 | 0.0002 | 0.0001 |
| BSG Flood | 0.9997 | 0.0002 | 0.0001 |
| SJK Flood | 0.9998 | 0.0001 | 0.0001 |

(2) Conditional probability of BL Overtopping induced by BL Flood (Table 3): The hydrological frequency method is employed to calculate the probability of the nodes. On the basis of observation data, the L-moment method is used to determine the statistical parameters, including mean $\bar{x}$, coefficient of variation ($C_v$), and coefficient of skewness ($C_s$). In most areas in China, P-III probability distribution is applicable to the floods. An acceptance–rejection sampling method was selected to calculate the conditional probability of overtopping induced by flood ($P(Overtoppin\ g/Flood)$). This method is determined using P-III type probability distribution.

$$x_i = b + \frac{1}{\beta}\left(-\sum_{k=1}^{[\alpha]}\ln\mu_k - B_i\ln\mu_i\right) \tag{7}$$

where $b = \bar{x}\left(1 - \frac{2C_v}{C_s}\right)$, $\beta = \frac{2}{\bar{x}C_vC_s}$, $\alpha = \frac{4}{C_s^2}$, and $i = 1,2,\cdots N$ As such, $N$ purely random members are generated responding to the P-III type distribution, $B = \frac{\mu_1^{1/r}}{\mu_1^{1/r} + \mu_2^{1/s}}$, $r = a - [a]$, $s = 1 - r$, $[a]$ is the integer, $\mu_1$, $\mu_2$ is a pair of random member.

A stochastic Monte Carlo simulation method was chosen to simulate the flood peak that exceeds the standard peak flow. The typical flood hydrograph of each dam was amplified by using the same magnification method to obtain the exceeding standard flood hydrographs. Finally, the highest water level of each dam was obtained according to the dam dispatching rules. After counting the number of times ($M$) the water levels exceed the dam crest, the conditional probability of dam overflow induced by flood can be calculated by:

$$P_f = P\left[(H - H_b) < 0\right] = M / N \tag{8}$$

where $P_f$ is the conditional probability of dam overflow induced by flood, $H$ is the crest height of the dam, $H_b$ is the highest water level of the dam, $M$ is the number of times that water levels exceed the dam crest, and $N$ is the number of simulations.

(3) Conditional probability of BSG Overtopping induced by BL Overtopping: In this study, two states of Wreck nodes are involved—occurrence and nonoccurrence. In general, an embankment dam breaks after flood overtopping. Therefore, the arcs between BSG Overtopping and BL Overtopping represent the dam breaking upstream.

**Table 3.** The control probability table of Bala (BL) Overtopping node.

| BL Overtopping | BL Flood | | |
|---|---|---|---|
| | Normal Flood | Check Flood | Overtopping Flood |
| occurrence | 0.00091 | 0.02 | 0.99909 |
| nonoccurrence | 0.99909 | 0.98 | 0.00091 |

**Table 4.** The control probability table of Busigou (BSG) Overtopping.

| BSG Overtopping | BL Overtopping | | | | | |
|---|---|---|---|---|---|---|
| | Occurrence | | | Nonoccurrence | | |
| BSG Flood | Normal Flood | Check Flood | Overtopping Flood | Normal Flood | Check Flood | Overtopping Flood |
| occurrence | 0.65 | 0.75 | 0.999999 | 0.000632 | 0.02 | 0.999368 |
| nonoccurrence | 0.35 | 0.25 | 0.000001 | 0.999368 | 0.98 | 0.000632 |

**Table 5.** The control probability table of Shuangjiangkou (SJK) Overtopping.

| SJK Overtopping | BSG Overtopping | | | | | |
|---|---|---|---|---|---|---|
| | Occurrence | | | Nonoccurrence | | |
| SJK Flood | Normal Flood | Check Flood | Overtopping Flood | Normal Flood | Check Flood | Overtopping Flood |
| occurrence | 0.45 | 0.55 | 0.999999 | 0.000035 | 0.02 | 0.999965 |
| nonoccurrence | 0.55 | 0.45 | 0.000001 | 0.999965 | 0.98 | 0.000035 |

The total capacities of BL and BSG are $1.38 \times 10^8$ m³ and $2.48 \times 10^8$ m³, respectively, so if the BL dam breaks due to a huge breaking flood from BL, the BSG dam will possibly break. Expert experience, dam capacity, discharge ability, dam type, and other factors are considered to determine the original conditional probabilities. If $P$ (BL Overtopping = occurrence) = 1, the occurrence probability of BSG Overtopping for three states of BSG Flood nodes is 0.65, 0.75, and 0.999999, as shown in Table 4. If $P$ (BL Overtopping = nonoccurrence) = 1, the probability of BSG Overtopping only depends on BSG flood. In the state of Overtopping flood, Equation (8) indicates that $P$ (BSG Overtopping = occurrence) = 0.999368; in the state of Normal flood, $P$ (BSG Overtopping = nonoccurrence) = $6.32 \times 10^{-4}$; expert experience is used to determine CPT in the check flood node. The conditional probability of SJK Overtopping is calculated in the same manner (Table 5).

As such, possible misperceptions of the probability of the event are generated. To avoid the subjective, CPT is updated after re-determining the continuous breaking failure paths.

The conditional probability in node C₃ is calculated as follows:

$$P\left(SJK_O\right) = \sum P\left(BL_F,\ BL_O,\ BSG_F,\ BSG_O,\ SJK_F,\ SJK_O\right)$$
$$= \sum_{B_O,\ C_F}\left\{P\left(SJK_O/BSG_O,\ SJK_F\right)\sum_{A_O,\ B}\left\{P\left(BSG_O/BL_O,\ BSG_F\right)\right\}\sum_{A_F}\left\{P\left(BL_O/BL_F\right)\right\}\right. \tag{9}$$

where $BL_F$ and $BL_O$ represent BL Flood and BL Overtopping, respectively, which is the same for BSG and SJK. In Formula (9), the conditional probability of the node SJK Overtopping contains the entire network's information and reflects the final evidence transfer result.

### 4.1.2. Sensitivity Analysis

In Hugin 8.5, the Parameter Sensitivity Analysis module was used to analyze the sensitivity of the hypothesis variable, which changes the value of the parameter variable and is appropriate for determining the risk factors for continuous breaking of cascade dams. In the sensitivity analysis, the node SJK Overtopping was selected as the hypothesis variable, while the other nodes were selected as parameter variables.

For the Flood node of each dam, the state of Overtopping was chosen to explain the results (Table 6). If SJK Overtopping = occurrence, the sensitivity of each node is 0.29, 0.45, and 1. The Flood factor with the highest sensitivity is SJK Overtopping Flood.

If the hypothesis variable SJK Overtopping = occurrence, the sensitivity of BL Overtopping = occurrence is $2.92 \times 10^{-5}$ (Table 7) in the Overtopping node of each dam. The sensitivity analysis for BSG Overtopping = occurrence has two cases: (1) If the parameter variable BL Overtopping = occurrence, the sensitivity of BSG Overtopping = occurrence is $4.56 \times 10^{-8}$; (2) if the parameter variable BL Overtopping = nonoccurrence, the sensitivity is $4.50 \times 10^{-5}$ (Table 7). That is to say, if BL

Overtopping = occurrence, BL Overtopping will be the sensitive factor; if BL Overtopping = nonoccurrence, BSG Overtopping will be the sensitive factor when SJK Overtopping = occurrence.

**Table 6.** The sensitivity analysis of SJK Overtopping, according to the Overtopping Flood node of each dam.

| SJK Overtopping | Overtopping Flood | | |
|---|---|---|---|
| Sensitivity Value | BL | BSG | SJK |
| occurrence | 0.29 | 0.45 | 1 |
| nonoccurrence | −0.29 | −0.45 | −1 |

**Table 7.** The sensitivity analysis of SJK Overtopping, according to the BL Overtopping node.

| SJK Overtopping | BL Overtopping | |
|---|---|---|
| Sensitivity Value | Occurrence | Nonoccurrence |
| occurrence | $2.92 \times 10^{-5}$ | $-2.92 \times 10^{-5}$ |
| nonoccurrence | $-2.92 \times 10^{-5}$ | $2.92 \times 10^{-5}$ |

**Table 8.** The sensitivity analysis of SJK Overtopping, according to the BSG Overtopping node.

| SJK Overtopping | BL Overtopping | | | |
|---|---|---|---|---|
| | Occurrence | | Nonoccurrence | |
| Sensitivity Value | BSG Overtopping | | | |
| | Occurrence | Nonoccurrence | Occurrence | Nonoccurrence |
| occurrence | $4.56 \times 10^{-8}$ | $-4.56 \times 10^{-8}$ | $4.50 \times 10^{-8}$ | $-4.50 \times 10^{-5}$ |
| nonoccurrence | $-4.56 \times 10^{-8}$ | $4.56 \times 10^{-8}$ | $-4.50 \times 10^{-5}$ | $4.50 \times 10^{-5}$ |

The Overtopping flood state of each dam is the most important sensitivity factor for dam overtopping. In this precondition, the failure path called "continuous breaking failure path 0" is identified in Figure 6. The risk transmission result in the original BN model is depicted in Figure 7. The probabilities of assumed to occur nodes are 100%, which showed as red, the posterior probability of the nodes are calculated by the cause–effect inference function of BNs, which showed as green.

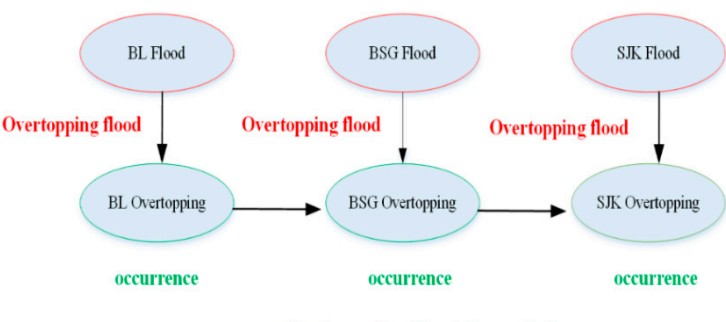

**Figure 6.** The continuous breaking failure path 0.

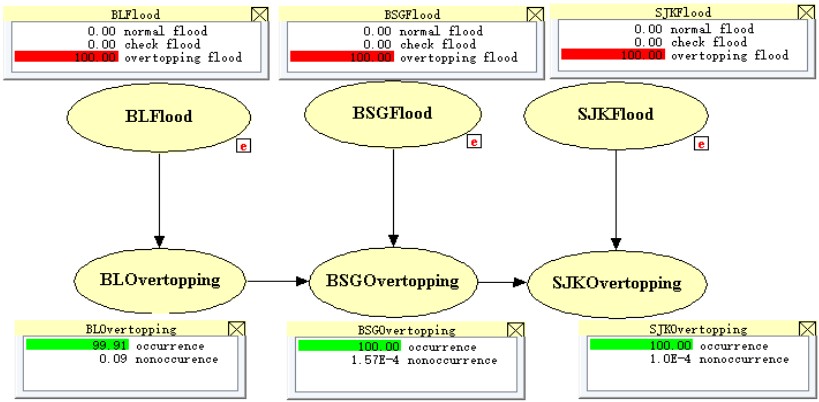

**Figure 7.** The inference result of continuous breaking failure path 0.

The cascade dam continuous breaking is a catastrophic event in one basin. Therefore, the details of the continuous breaking progress needed to be explicitly analyzed.

### 4.2. Dam Breaking and Flood Routing Analysis

#### 4.2.1. Dam Continuous Breaking Analysis for Situation 1

Two problems are specifically analyzed in this section: The flood hydrograph of dam breaking and the downstream breaking flood route causing downstream dam breaking.

In the continuous breaking failure path 0, the first potential factor triggering risks is the BL Overtopping flood. The hydrology frequency and the same ratio amplification method were applied to determine the overtopping flood.

During the analysis, the flood regulation method in Equation (10) and "water lever volume" (Z–V), and "water level discharge" (Z–Q) of each dam are applied whenever the water level rises above the dam crest:

$$\frac{Q_1 + Q_2}{2}\Delta t - \frac{q_1 + q_2}{2}\Delta t = V_2 - V_1 \tag{10}$$

where $\Delta t$ is the time step; $Q_1$ and $Q_2$ are the initial and final inflows, respectively; $q_1$ and $q_2$ are the initial and finial outflows, respectively; and $V_1$ and $V_2$ are the initial and final volumes, respectively.

If the water level reaches crest elevation, the embankment dam begins to break, and then the breaking process is simulated by DB-IWHR 2014.

As shown in Figure 8, the BL water level reaches the crest and BL begins to break at 18.6 h among the flood routing results of the BL overtopping flood. The breaking flood routes to SJK after three hours. As the BL–BSG–SJK cascade dam system is a narrow valley, the flood peak and flood hydrograph routing to the downstream are the same as the breaking hydrograph. The BL breaking flood routes to the SJK dam site is depicted in Figure 9A.

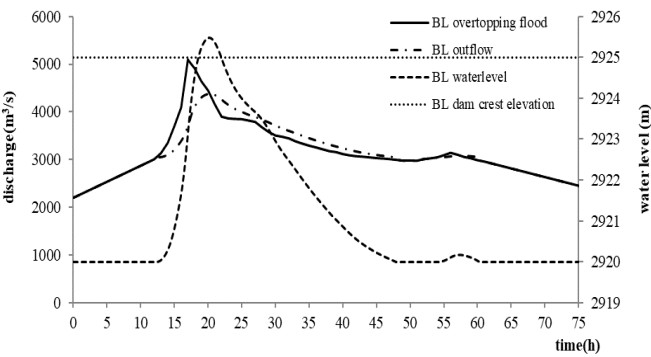

**Figure 8.** The regulated flood progress of BL (overtopping flood).

To analyze dam breaking floods, some parameters need to be set in DB-IWHR 2014. In addition, the breaking parameters of BL and BSG are listed in Table 9, according to the engineering design materials of each dam and previous studies [19,26,28] related to embankment dam breaking events in China.

**Table 9.** The parameters for dam break analysis case.

| Item | Parameter | BL Dam Break Parameters Value | BSG Dam Break Parameter Value Under BL Break |
|---|---|---|---|
| Water level–volume relationship | $a_1$ | 0.03 | 0.12 |
| | $b_1$ | 3.43 | 5.52 |
| | $c_1$ | 121.00 | 228.40 |
| Elevation of dead water | $H_r$ | 2918.00 m | 2600.00 m |
| Erosion rate | $V_c$ | 3.00 m³/s | 3.00 m³/s |
| | $a_2$ | 1.1000 | 1.1000 |
| | $b_2$ | 0.0010 | 0.0010 |
| Lateral enlargement | $B_0$ | 10.00 m | 70.00 m |
| | $B_{end}$ | 60.00 m | 192.00 m |
| | $\alpha$ | 155° | 90° |
| | $\beta$ | 165° | 90° |
| | $Z_0$ | 2925.00 m | 2600.00 m |
| | $Z_{end}$ | 2900.00 m | 2547.00 m |

Note: The water level–volume relationship: $W = a_1 \left( H - H_r \right)^2 + b_1 \left( H - H_r \right) + c_1$. $V_c$ is the incipient velocity, $1/a$ is the tangent of this curve at the incipient stress, $1/b$ is the maximum possible erosion ratio, $Z_0$ is the incipient elevation of the channel bed, and $Z_{end}$ is the end elevation of the channel bed, $B_0$ is the initial breach width, $B_{end}$ is the end breach width.

The worst hydrologic engineering situation was considered in this study: BL breaking flood encounters BSG overtopping flood. The inflow of the downstream dam is composed of the upstream breaking flood and the downstream natural flood. For the single flood peak type, the breaking flood peak is superimposed on the natural flood peak, during the other time step, the flood volume corresponding superposition. For the double flood peak type, the breaking flood peak is superimposed at the middle time of the two flood peaks of the natural flood, during the other time step, the flood volume corresponding superposition. As shown in Figure 9B, the BSG overtopping flood is superimposed on the BL breaking flood to determine the inflow at BSG.

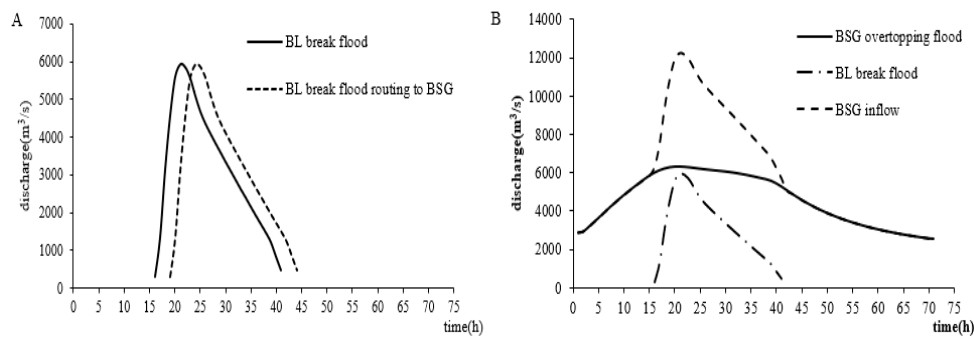

**Figure 9.** (**A**) The BL break flood routing to SJK dam site. (**B**) The BL break flood encounters BSG overtopping flood.

The regulated flood progress was computed with Equation (10) and the Z–Q and Z–V of the BSG dam. Through analyzing the performance results in Figure 10, the BSG dam will begin to break if the BSG water level attains 2608 m. The breaking analysis parameters are explained in Table 9. The BSG breaking flood is shown in Figure 11A on the basis of breaking progress analysis. If the flood peak is 21,877.13 m³/s, a huge released flood will route to the downstream dam site and then superimpose on SJK overtopping flood, with the largest inflow of SJK attaining 30,410 m³/s. However, the magnitude of inflow still cannot be contained by the SJK dam. The conceptional regulated flood progress is depicted in Figure 11B, and the highest regulation water level remains at 2518.65 m.

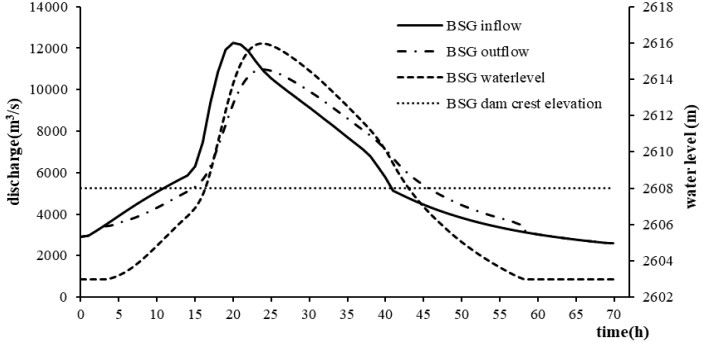

**Figure 10.** The BSG regulation progress.

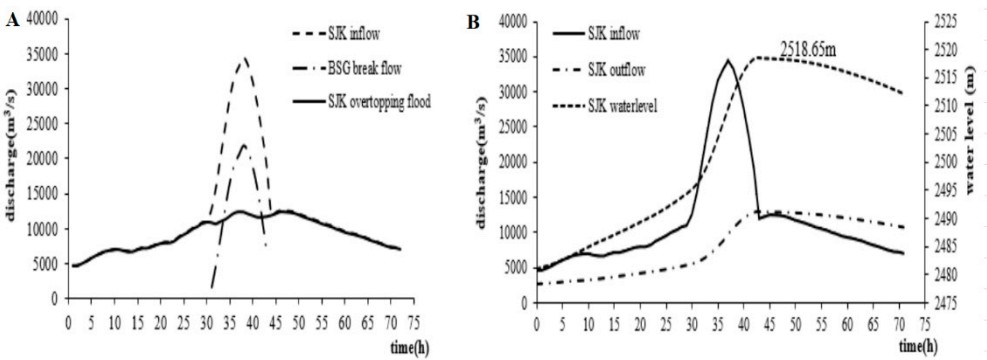

**Figure 11.** (**A**) The BSG break flood encounters SJK overtopping flood; (**B**) the regulated flood progress of SJK.

When analyzing situation 1, we found that the largest inflows of BSG and SJK were 12,249.49 m³/s and 30,410 m³/s, respectively. In addition, extreme breaking flow can be generated in the

channel in this situation, and inestimable catastrophe and damage will be caused downstream since all three dams break. However, it was also important to research the continuous breaking events, due to check flood and normal flood when considering the frequency of the overtopping flood.

If new evidence is found in the continuous breaking failure paths, the original CPTs of the BNs model should be updated under the new conditions.

### 4.2.2. Dam Continuous Breaking Analysis for Situation 2

To analyze the continuous breaking conditions of the cascade dam in the flowing situation, the first factor triggering risks were set to $P$ (BL Overtopping = occurrence) = 1, the same as situation 1. The released flood routes to the BSG dam site after BL dam breaking and the BSG check flood was chosen to encounter the BL break flood in this situation. As shown in Figure 12A, the BSG inflow is composed of BL break flood and BSG check flood. As shown in Figure 12B, BSG begins to break if the water level reaches crest elevation at 18.5 h during the BSG regulated flood progress. If the BSG break flood superimposes on the SJK check flood, the flood peak will be 26,520.67 m³/s, and the SJK dam will also begin to break at 38.7 h because of this inflow.

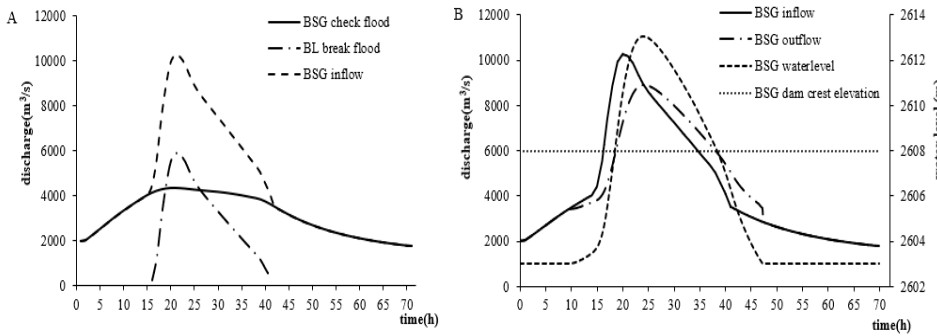

**Figure 12.** (**A**) The BL break flood encounters BSG check flood; (**B**) the regulated flood progress of BSG.

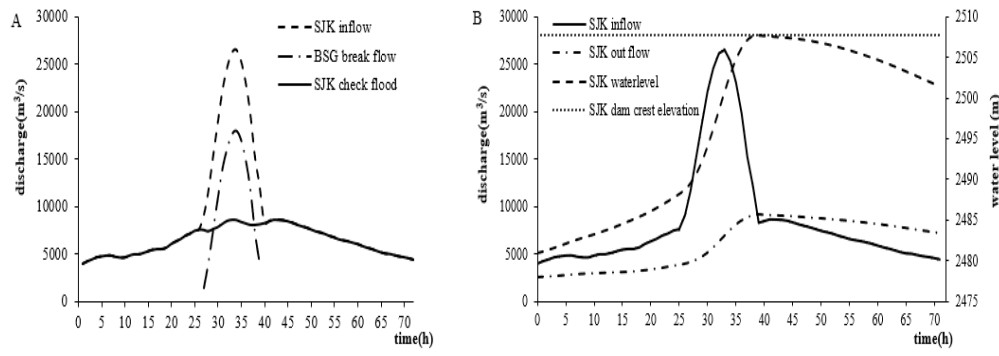

**Figure 13.** (**A**) The BSG break flood encounters SJK check flood; (**B**) the regulated flood progress of SJK.

When the break flow encounters the BSG check flow, the BSG dam break will be caused by the composed flood, different from situation 1. If the BSG break flood reaches its peak of 17,910.67 m³/s, the BSG break flood will superimpose on the SJK check flood and SJK will attain the dam crest of 2507.7 m at 38.7 h, at which time the SJK dam will start breaking.

The normal flood of SJK was selected as flood progress in the typical year of 1981 (Figure 14). SJK normal flood was chosen to superimpose on the BSG breaking flood (Figure 15A). The regulated results with the entering of the combined flood into the SJK dam are shown in Figure 15B, as the SJK dam is safe if the maximum water level is 2500.91 m.

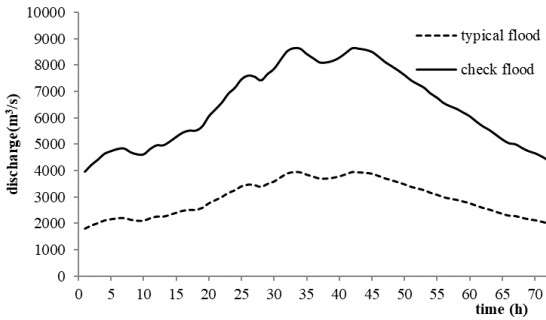

**Figure 14.** SJK dam check flood hydrographs and measured flood hydrograph in 1981.

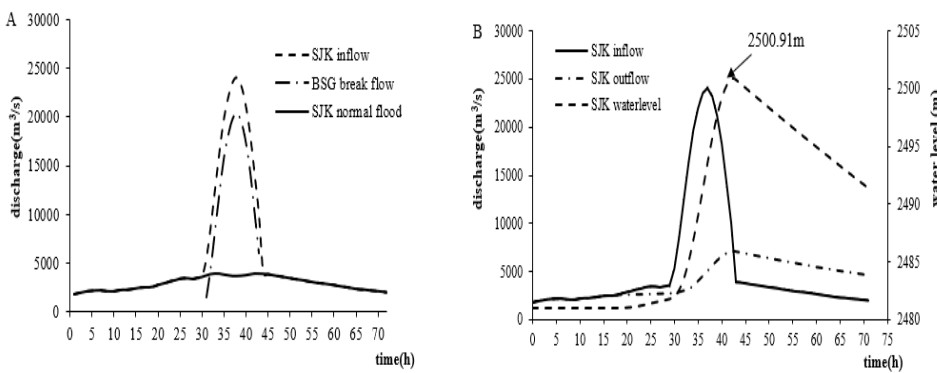

**Figure 15.** (**A**) The BSG break flood encounters SJK normal flood; (**B**) the regulated flood progress of SJK.

For situation 2, the continuous breaking failure path can be divided into two types: The BL–BSG–SJK cascade dam system continuous breaking, and the BL–BSG continuous breaking, during which the SJK dam is safe.

### 4.2.3. Dam Continuous Breaking Analysis for Situation 3

As for situation 3, the BL breaking flood is composed of BSG typical flood (the typical year in 1981), which enables the BSG dam to attain the dam crest of 2608 m at 19.1 h (Figure 16B), at which time the BSG dam begins to break. In addition, the peak of the BSG breaking flood is 16,681.53 m³/s (Figure 17A), at which time the BSG breaking flood is superimposed on the SJK typical flood. After being regulated by SJK (Figure 17B), the SJK dam is safe if the maximum water level reaches 2496.57 m.

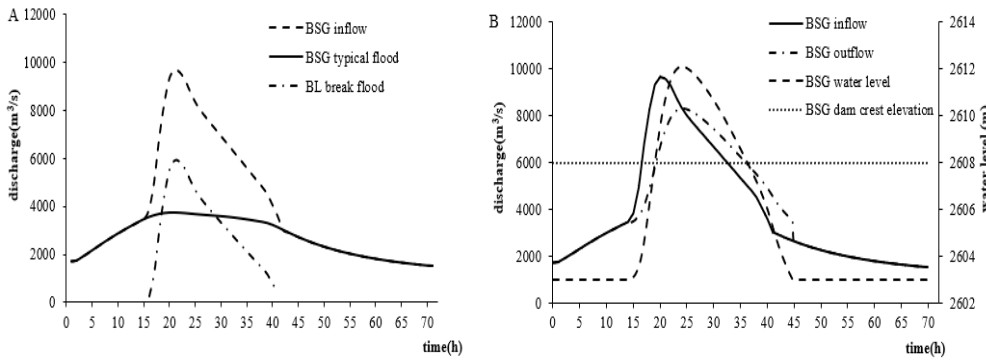

**Figure 16.** (**A**) The BL break flood encounters BSG typical flood; (**B**) the regulated flood progress of BGS.

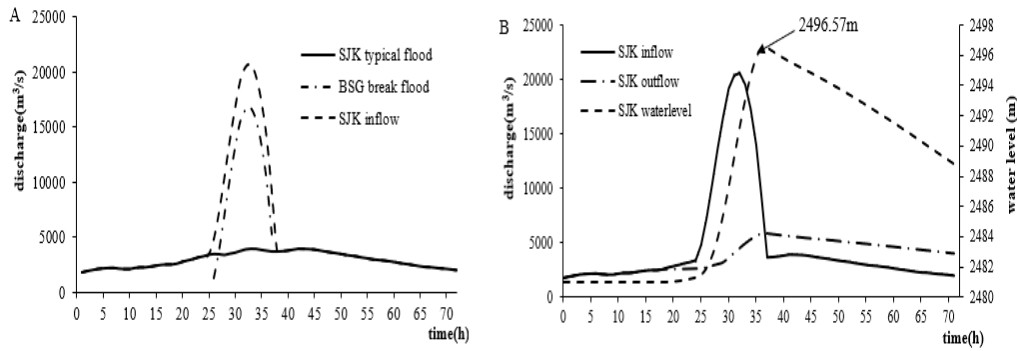

**Figure 17.** (**A**) The BSG break flood encounters SJK typical flood; (**B**) the regulated flood progress of SJK.

The above analysis shows that the BL and BSG dams continuously break, while the SJK dam is safe when BSG breaking flood routes to the SJK dam site and superimposes on the SJK typical flood. To ensure the reliability of the results, the SJK check flood was also chosen to be combined with the BSG breaking flood. As shown in Figure 18B, the SJK regulated results show that the SJK dam is safe if the maximum water level is 2505.92 m. As for situation 3, under the breaking of BL, BSG breaks when there is one typical flood, whereas the SJK dam is safe whenever there is a normal or check flood.

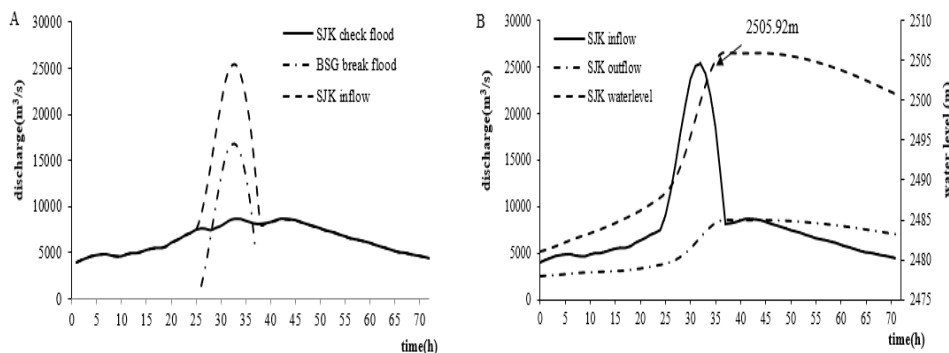

**Figure 18.** (**A**) The BSG break flood encounters SJK check flood; (**B**) the regulated flood progress of SJK.

### 4.3. CPTs Updating Based on the Continuous Breaking Path Analysis

The cause–effect relationships of the BN model are represented by CPTs, which can determine the system probability transmission. We found that the CPTs are updated as flowing, through analyzing the continuous breaking conditions.

As explained in Table 10, if $P$ (BL Overtopping = occurrence) = 1, the probabilities of $P$ (BSG Overtopping = occurrence) are 0.9, 0.99, and 0.999, respectively, corresponding to three flood states of normal, check, and overtopping. If $P$ (BL Overtopping = nonoccurrence) = 1, the probabilities of $P$ (BSG Overtopping = occurrence) only depend on its own failure probabilities.

**Table 10.** The probability of BSG Overtopping on the condition of BL Overtopping and BSG Flood.

| BSG Overtopping | BL Overtopping | | | | | |
|---|---|---|---|---|---|---|
| | Occurrence | | | Nonoccurrence | | |
| BSG Flood | Normal Flood | Check Flood | Overtopping Flood | Normal Flood | Check Flood | Overtopping Flood |
| occurrence | 0.9 | 0.99 | 0.999 | 0.000632 | 0.02 | 0.999368 |
| nonoccurrence | 0.1 | 0.01 | 0.001 | 0.999368 | 0.98 | 0.000632 |

**Table 11.** The probability of SJK Overtopping on the condition of BSG Overtopping and SJK Flood.

| SJK Overtopping | BSG Overtopping | | | | | |
|---|---|---|---|---|---|---|
| | Occurrence | | | Nonoccurrence | | |
| SJK Flood | Normal Flood | Check Flood | Overtopping Flood | Normal Flood | Check Flood | Overtopping Flood |
| occurrence | 0.25 | 0.99 | 0.999999 | 0.000035 | 0.02 | 0.999965 |
| nonoccurrence | 0.75 | 0.01 | 0.000001 | 0.999965 | 0.98 | 0.000035 |

According to situations 2 and 3, the probabilities of SJK Overtopping under the condition of BSG Overtopping and SJK Flood should be divided into two types.

For situation 2, if $P$ (BSG Overtopping = occurrence) = 1, the SJK dam is safe when there is a normal flood and will break when there is a check flood. This research emphasizes the flood factor that leads to the dam breaking. However, dam breaking, such as landslide and piping, can be caused by other factors. Considering this and expert experiments carefully, we found that the conditional probabilities $P$ (SJK Overtopping = occurrence) are 0.25 and 0.99, respectively. The other nodes of conditional probabilities are the same as those in Table 5.

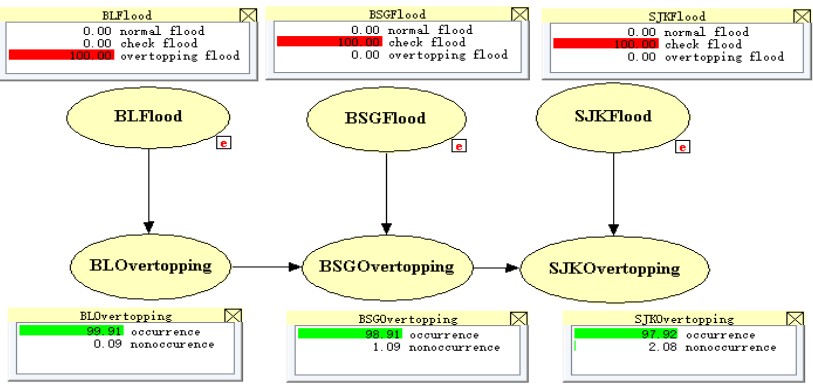

**Figure 19.** The posterior probability of each dam's overtopping node on the condition of Flood evidence.

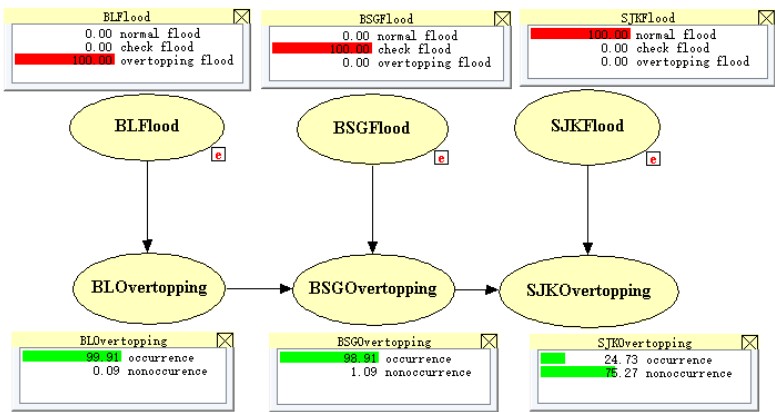

**Figure 20.** The posterior probability of each dam's overtopping node on the condition of Flood evidence.

After updating the CPTs of BSG Overtopping and SJK Overtopping, Figures 19 and 20 show that the posterior probabilities of each dam also changed. For situation 2, there are two continuous breaking paths: One is BL–BSG–SJK continuous breaking, with posterior probabilities of each dam of 0.9991, 0.9891, and 0.9792, respectively; and the other is BL–BSG continuous breaking, with the posterior probabilities of 0.9991, 0.9891, and 0.2473, respectively.

For situation 3, BL and BSG dams break continuously, and the SJK dam is safe when normal or check floods occur. As shown in Table 12, the probabilities of *P* (SJK Overtopping = occurrence) are 0.25 and 0.35 when SJK normal and check floods occur, respectively, under the condition of the updated CPTs of SJK Overtopping.

**Table 12.** The probability of SJK Overtopping under the condition of BSG Overtopping and SJK Flood.

| SJK Overtopping | BSG Overtopping | | | | | |
|---|---|---|---|---|---|---|
| | Occurrence | | | Nonoccurrence | | |
| SJK Flood | Normal Flood | Check Flood | Overtopping Flood | Normal Flood | Check Flood | Overtopping Flood |
| occurrence | 0.25 | 0.35 | 0.999 | 0.000035 | 0.02 | 0.999965 |
| nonoccurrence | 0.75 | 0.65 | 0.001 | 0.999965 | 0.98 | 0.000035 |

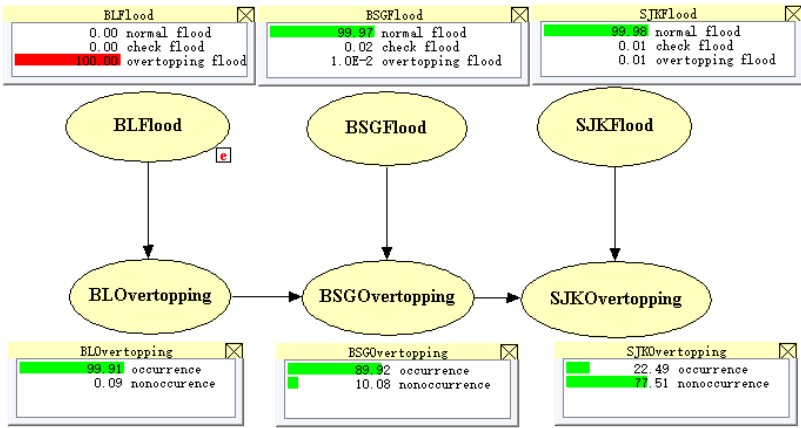

**Figure 21.** The posterior probability of each dam's overtopping node on the condition of Flood evidence.

Based on the updated conditional probability in Table 12, the posterior probabilities of each dam's Overtopping node are shown in Figure 21. The overtopping posterior probabilities of BL, BSG, and SJK are 0.9991, 0.8992, and 0.2249, respectively.

## 4.4. Cascade Dam System Failure Probability

Three continuous break failure paths for the cascade BL–BSG–SJK were elicited by the dam break and flood routing analysis in Section 4.2. The first continuous failure path is shown in Figure 22A: BL-BSG-SJK cascade sequential breaking occurs if the flood states of BL, BSG, and SJK are Overtopping, Check, and Check flood, respectively. The second continuous failure path is shown in Figure 22B: BL and BSG break, while SJK is safe if the flood states of BL, BSG, and SJK are Overtopping, Check, and Normal flood, respectively. The third failure path is depicted in Figure 22C, where the flood states of each dam are BL Flood = Overtopping flood, BSG Flood = Normal flood, and SJK Flood = Normal/Check flood.

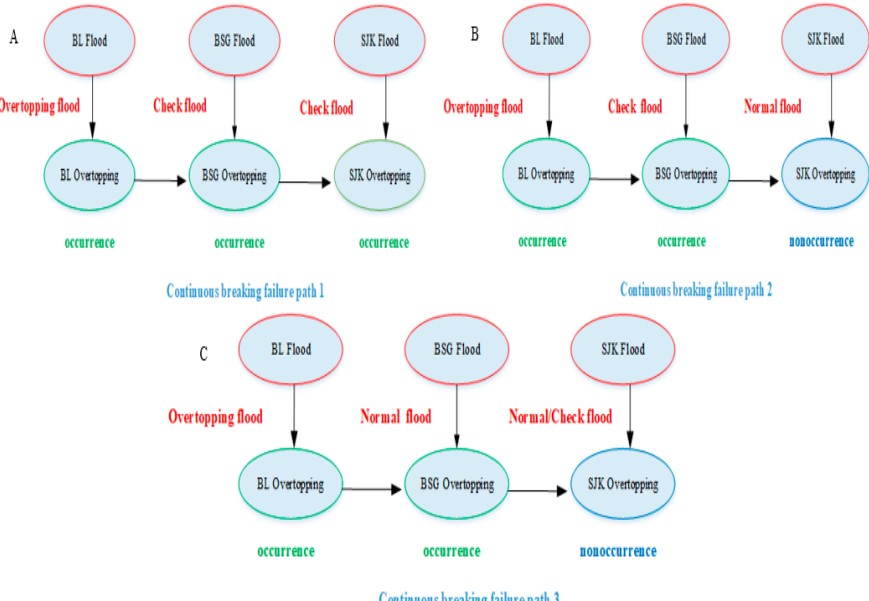

**Figure 22.** (**A**) The continuous breaking failure path 1, (**B**) the continuous breaking failure path 2, and (**C**) the continuous breaking failure path 3.

After identifying the continuous breaking failure paths, the system failure probabilities of the failure paths were computed by the BNs model. The system failure probability is 1 in the continuous breaking failure paths 0 and 1, since all three dams break. The system failure probabilities of continuous breaking failure paths 2 and 3 were determined as flowing.

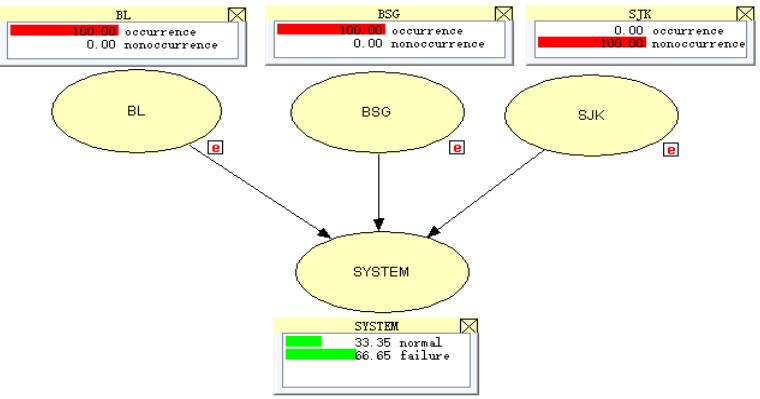

**Figure 23.** The cascade system failure probability of failure path 2.

As shown in Figure 23, the calculated model of system failure probability was established with a focus on the impacts of every dam on the cascade dam system. The prior probability of each dam was obtained with the consideration of the failure path states and the CPTs (Tables 4 and 5) in this model. For instance, check flood occurs at the BSG dam and the upstream BL dam is overtopped in this path; thus, the prior probability of BSG Overtopping = occurrence is 0.75, which can be obtained in Table 4. According to continuous breaking path 2, the evidence is set as $P$ (BL Overtopping = occurrence) = 1, $P$ (BSG Overtopping = occurrence) = 1, and $P$ (SJK Overtopping = nonoccurrence) = 1, and then the system failure probability is computed as $P$ (system = normal) = 0.3335 and $P$ (system = failure) = 0.6665.

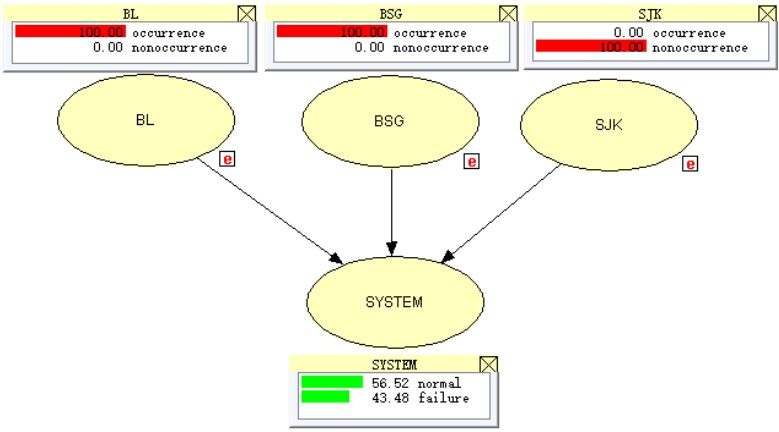

**Figure 24.** The cascade system failure probability of failure path 3.

The system failure probability based on continuous breaking path 3 is shown in Figure 24. In this model, the prior probability of the nodes is based on the identification results of failure path 3. For instance, the BSG dam breaks when BL breaking and normal flood occur, resulting in the prior probability $P$ (BSG Overtopping = occurrence) = 0.65, which can be derived from Table 4. Based on continuous breaking path 3, the evidence in this model is set as $P$ (BL Overtopping = occurrence) = 1, $P$ (BSG Overtopping = occurrence) = 1, and $P$ (SJK Overtopping = nonoccurrence) = 1; and the system failure probability is finally calculated as $P$ (system = normal) = 0.5652 and $P$ (system = failure) = 0.4348.

## 5. Conclusions

Considering the characteristics of a cascade dam system, BNs and DB-IWHR 2014 were combined in this study to analyze the flood risks of a cascade dam system. The method was applied to the BL–BSG–SJK cascade dam system in the Dadu River basin in China.

On the basis of the inference and the information-transmitting functions of BNs, the original sequential breaking path of the cascade dam system was confirmed using sensitivity analysis. However, some errors in the results occur since the original BN model is heavily decided by historical data and experts' experience. To overcome this limitation, DB-IWHR 2014 and the flood regulation method were applied to simulate the breaking process. After analyzing the breaking simulations, the more reasonable continuous breaking failure paths were determined, such as the first: BL Flood = overtopping, BSG Flood = check flood, and SJK Flood = check flood. BL, BSG, and SJK sequentially break under this situation. The second path is BL Flood = overtopping, BSG Flood = check flood, and SJK Flood = normal flood; BL and BSG break, while SJK is safe under this situation. The third path is BL Flood = overtopping, BSG Flood = normal flood, and SJK Flood = normal/check flood. Here, BL and BSG break, while SJK is safe, which is the same as for path two. Through analyzing the three paths, we found that continuous breaking events occur in both overtopping flood and in check/normal flood of the cascade dam system. For this reason, the CPTs of the original BNs model should be updated by adding the new information. Finally, a new BN model was established to quantify the failure probabilities of the cascade system under the three continuous breaking failure paths.

The subjectivity can be reduced and the reliability of flood risk analysis of cascade dam systems can be enhanced with the combination of BNs and DB-IWHR 2014. The results can provide some suggestions to decision-makers. If the BL dam overtopping happened, according to the return period of natural floods, proper flood control measurements should be provided to avoid a cascade of continuous dam breaks. This method can be easily adapted to other cascade dam systems.

**Author Contributions:** W.C. and X.Z. defined the research themes and designed the methods and modeling. W.C. and A.P. analyzed the data. X.W. and Z.F. proofread the language and gave some advice for the revised manuscript. All authors have contributed to the revision and approved the manuscript.

**Funding:** This research received no external funding.

**Conflicts of Interest**: The authors declare no conflict of interest.

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
