# Peer review of "Flood Risk Analysis for Cascade Dam Systems: A Case Study in the Dadu River Basin in China"

_water, doi:10.3390/w11071365_

Round 1
Reviewer 1 Report
Dear Authors,
in my opinion the manuscript does not meet the minimum requirements to be published as a scientific paper.
My concerns/suggestions are presented herein (in a non-specific importance order):
1 - There are a lot of english language problems in the text. I would recommend a complete revision of the manuscript. Examples: line 38 - 40 needs to be rewritten and line 72 "Then the results are able to determine the continuous failure path for cascade dam system."
2 -One example it is not clear what the authors wanted to say: in line 59: "In many studies, ..."
3 - Very good overview for the dam construction in China (1st paragraph of introduction).
4 - In the study area section, more details regarding the dams may be given. Do these dams have the same construction type? How was the maximum discharge estimated for each dam i.e. what is the return period? Were there any accidents before?
5 - in this river there are another dams, why weren't they included in the analysis?
6 - In this case, the triggering factors are only due to overtopping. How can other factors like seismic of piping be included in your analysis?
7 - Line 114, t and z are not introduced.
8 - The original probabilities are estimated in expert experience? In this case, the authors should put the focus on the procedure instead of the calculated probabilities.
9 - The situations referred in section 4.2 are not presented before.
10 - Please give main highlight of the term incipient continuous failure path (e.g. line 48).
Author Response
Dear reviewer.
We are very glade to receive your comments, we provide a point-by-point response to the comments and revised all of them in the revised manuscript, the responses and revised manuscript was uploaded as Word files.

Reviewer 2 Report
Attached please find my suggestions.

Author Response

(The authors gave the same response as above.)

Round 2
Reviewer 1 Report
The suggestions were considered in this revised version